# COVID-19 Vaccine Hesitancy: A Global Public Health and Risk Modelling Framework Using an Environmental Deep Neural Network, Sentiment Classification with Text Mining and Emotional Reactions from COVID-19 Vaccination Tweets

**DOI:** 10.3390/ijerph20105803

**Published:** 2023-05-12

**Authors:** Miftahul Qorib, Timothy Oladunni, Max Denis, Esther Ososanya, Paul Cotae

**Affiliations:** 1Department of Computer Science and Information Technology, University of the District of Columbia, Washington, DC 20008, USA; 2Department of Mathematics and Statistics, University of the District of Columbia, Washington, DC 20008, USA; 3Department of Computer Science, Morgan State University, Baltimore, MD 21251, USA; 4Department of Mechanical and Biomedical Engineering, University of the District of Columbia, Washington, DC 20008, USA; 5Department of Electrical and Computer Engineering, University of the District of Columbia, Washington, DC 20008, USA

**Keywords:** COVID-19, public health vaccine hesitancy, emotion, sentiment risk analysis, environmental neural network modeling, deep learning, Twitter

## Abstract

Popular social media platforms, such as Twitter, have become an excellent source of information with their swift information dissemination. Individuals with different backgrounds convey their opinions through social media platforms. Consequently, these platforms have become a profound instrument for collecting enormous datasets. We believe that compiling, organizing, exploring, and analyzing data from social media platforms, such as Twitter, can offer various perspectives to public health organizations and decision makers in identifying factors that contribute to vaccine hesitancy. In this study, public tweets were downloaded daily from Tweeter using the Tweeter API. Before performing computation, the tweets were preprocessed and labeled. Vocabulary normalization was based on stemming and lemmatization. The NRCLexicon technique was deployed to convert the tweets into ten classes: positive sentiment, negative sentiment, and eight basic emotions (joy, trust, fear, surprise, anticipation, anger, disgust, and sadness). *t*-test was used to check the statistical significance of the relationships among the basic emotions. Our analysis shows that the *p*-values of joy–sadness, trust–disgust, fear–anger, surprise–anticipation, and negative–positive relations are close to zero. Finally, neural network architectures, including 1DCNN, LSTM, Multiple-Layer Perceptron, and BERT, were trained and tested in a COVID-19 multi-classification of sentiments and emotions (positive, negative, joy, sadness, trust, disgust, fear, anger, surprise, and anticipation). Our experiment attained an accuracy of 88.6% for 1DCNN at 1744 s, 89.93% accuracy for LSTM at 27,597 s, while MLP achieved an accuracy of 84.78% at 203 s. The study results show that the BERT model performed the best, with an accuracy of 96.71% at 8429 s.

## 1. Background

In 2019, in Wuhan, China, coronavirus disease 2019 (COVID-19) was reported for the first time [1,2], and it later spread rapidly to 72 countries [3] then started to deteriorate children’s learning process [4]. Before the Chinese government quarantined Wuhan (on 23 January 2020), it was estimated that COVID-19 was spread to 369 cities in China [5]. The COVID-19 pandemic which started at the end of 2019 remained unbated in 2021. Coronavirus disease 2019 (COVID-19), which started at the end of 2019 and had spread worldwide [3], remained persistent in 2021. In April 2020, at the peak of school closures, there were about 1.6 billion children out of school worldwide, and around 700 million kids were studying from home due to the huge uncertainty between schools and families about having hybrid and remote learning, or no school at all [6]. Thus, for almost two years, COVID-19 has disrupted the education system globally [7] and impacted vulnerable learners enormously. School closures, lack of connectivity, and limited devices caused one-third of students to have difficulties pursuing learning virtually.

The rate of depression and anxiety associated with COVID-19 has increased dramatically [8]. A report showed that 14.4% of high schoolers have stress disorders, while 40.4% have anxiety and depression [9]. Life satisfaction for teenagers has suffered an enormous decrease from 92% to 72% because of school lockdown [10]. Many higher education institutions have suffered tremendously from COVID-19. Historically Black Colleges and Universities (HBCUs), Tribal Colleges and Universities (TCUs), and Minority Serving Institutions (MSIs) declined their enrollments during the academic year of 2020 to 2021 [11]. The U.S. Department of Education report (2021) also states that there was a sharp drop-off in high school graduates, especially among high-poverty schoolers, compared to the prior pre-pandemic years.

COVID-19 has also changed the socioeconomic structures of societies. The pandemic has increased anxiety and depression due to the changes in the teaching–learning process from traditional to remote, which has widened socioeconomic gaps [8]. As a result, there is no surprise that medical personnel and other professions have been severely affected and required to make extreme changes to deal with the pandemic. Many families were severely impacted by the loss of wages (some of the losses due to parents being required to stay home with children), which led to food and housing insecurity [12]. It has been documented that economic instability leads to an increase in depression and anxiety [13]. A report shows that 37.71% of parents noticed behavioral problems in their children because of the pandemic and virtual educational system [14].

The pandemic has not only created socio-economic chaos in education and business, but it also has behavioral, emotional, and psychological impact on communities, including psychological disorders, fear, depression, anxiety, and suicidal ideation [15,16]. In addition, the pandemic has created multiple psychological stresses associated with an increased risk of mental illness [17]. Compounding effects of the novel virus, such as economic downtime, negatively affected people by aggravating conditions such as worry, stress, mental illness, anxiety, and depressive disorders [18]. The COVID-19 restriction worsened individuals’ emotions and worries for children and adolescents with or without mental health conditions [19]. Due to the large closures, underdeveloped, developing, and advanced countries were severely obstructed by the pandemic [20]. Governments provided massive financial support to protect businesses, households, and vulnerable communities; thus, reallocating public funding to support priorities, such as healthcare, was very crucial [21]. At the end of November 2020, various pharmaceutical corporations proclaimed an innovation of the COVID-19 vaccine [22].

The Maryland Emergency Management Agency gathered various vaccine rumors that were spread in societies [23]. The first rumor is that vaccination will infect people with COVID-19. Another speculation is that vaccination will alter people’s deoxyribonucleic acid (DNA) because messenger RNA vaccines (mRNA) introduce a protein that triggers an immune response. A third rumor is that such vaccine is not safe since it was quickly developed and insufficiently tested. Next, there is gossip that there is a microchip to track people in such vaccine. The other anti-vaccination rumor is that COVID-19 vaccines will affect the ability to have a baby and are harmful to pregnant women and babies who are breastfed by vaccinated women.

Unfortunately, there are individuals and groups who refuse to accept vaccines to combat the coronavirus disease and oppose the use of vaccines due to rumors, myths, anti-vaccination campaigns, and conspiracy theories [24]. Some of the reasons consistently spread by anti-vaccine groups are related to vaccine development process and long-term effects [25]. Consequently, many researchers and public health experts have been continuously gathering dataset to have a better understanding on vaccine hesitancy [26].

Nowadays, the Internet has been expanding its use to provide information related to health [27]. Many researchers gather rich and free information on COVID-19 vaccination in societies from the Internet [28]. Social media has been an astonishing source of information, where societies continuously express their beliefs through social networks [28,29]. Furthermore, the application of text mining to collect datasets from social media has been a meaningful tool [30], which makes it possible that text mining can be an alternative method to analyze public opinions on COVID-19 vaccine hesitancy [31]. In addition, social media have also played a major role in educating anti-vaccine groups, and traditional channels, such as national television and national and local newspapers, continuously promote vaccine information to their audiences [32]. In this study, extracted emotions from COVID-19 datasets were gathered from social media for sentiment analysis. 

This paper is organized as follows: Section 2 reviews current literature utilizing deep learning to understand COVID-19 hesitancy, Section 3 explains the methodology of the research study, and Section 4 describes the experimental results. The conclusion is presented in Section 5. Future work is presented in Section 6, and acknowledgments are presented in Section 7.

## 2. Literature Review

### 2.1. Previous Studies

There have been various studies on COVID-19 vaccine hesitancy utilizing text-based sentiment analysis since the outbreak of the pandemic in 2019. Many previous studies used machine learning techniques to analyze vaccine hesitancy based on Twitter datasets [33,34,35,36,37]. Some researchers applied a deep learning approach to conduct text-based sentiment analysis on social media platforms [38,39,40], while other researchers used multiple techniques combining machine and deep learning approaches to analyze public opinions on COVID-19 based on social media datasets, such as from Twitter, to achieve better model performance [41,42,43,44,45].

Assessing the use of supervised learning in sentiment analysis [41] expounds on the need for identifying sentiments as a guide to better decision making due to the presence of subjective opinions and biases in reporting on COVID-19. This work tries to answer questions on how well supervised learning can perform in sentiment analysis through feature engineering. The authors of [41], using five classifiers, demonstrated how machine learning can be used for methodical investigation of Twitter sentiments expressed by social media users by using Radom Forest, XGBoost, and Support Vector classifiers. In their study, a comparative approach of deep learning is suggested to evaluate the performance of COVID-19 Twitter sentiment analysis, especially in addressing the complexity associated with the decidability of supervised learning.

In related literature, the impact of divergence support on COVID-19 vaccination is also assessed. Sentiment analysis of online platforms has emerged as one of the best indicators in getting an idea of what’s on people’s minds and, subsequently, what may be trending due to public opinions on an issue that may be contested. The authors of [38] used deep learning methods to analyze COVID-19 vaccination responses and demonstrated the use of Natural Language Processing tools for the predictive performance of RNN, LSTM, and Bi-LSTM in understanding public opinions. The author showed the importance of sentiment analysis in measuring a critical parameter (vaccination), which by far is known to be most impactful in the fight against COVID-19. The aim of this study was to understand the dominant feelings before and after vaccination to create modified approach. This study, among other comparative studies, demonstrates the leverage of sentiment analysis, especially in the application of decision science.

Sentiment analysis on COVID-19 based on geographical location and setting has also been studied. Liu’s examination [46] used geolocation to compare online COVID-19 Twitter sentiments between urban and rural residents. In the cluster analysis which used trained word2vec models, urban and rural sentiments based on hashtags and opinion objectives were also investigated in the study. The inference of the study, which deployed geospatial and timing information, found a significant statistical contrast in sentiments for urban and rural Twitter users. The study results demonstrated the leverage of geographically targeted measures in epidemic prevention.

The use of deep learning for online sentiment analysis is also expounded by Hernandez [47] in a performance comparison of multilingual BERT pre-trained models. This study on the topic of COVID-19 in Mexico used Spanish-language parameters to compare pretrained Spanish and multilingual BERT models against other machine learning algorithms, such as Naïve Bayes, SV, Logistic Regression, and Decision Trees. In the paper, the authors presented the sentiments of the Mexican population on COVID-19 using lexical-based data labeling and a semi-supervised technique. Their work demonstrates the use of bi-directional transformers and how efficiently the method can classify tweets with respect both positive and negative sentiments. However, this study also acknowledges the need for models that can recognize semantics such as irony and sarcasm, and words that have different meanings, since this would be a challenge while classifying contrary opinions. 

Two popular sentiment methods to compute text-based sentiment analysis in these works are TextBlob and VADER. Some research studies used TextBlob sentiment computation [41,48] to investigate COVID-19 vaccine hesitancy using Twitter datasets. Other research studies [49,50] applied VADER to examine vaccine hesitancy.

Table 1 represents a comparative study of text-based sentiment analysis to understand COVID-19 vaccine hesitancy based on Twitter datasets and utilizing deep learning approach. Some of the projects used deep learning techniques only, while others combined machines and deep learning methods. Comparing prior studies enhances our research in analyzing various areas that have not been considered in these prior studies. Table 1 is a representation of various studies with different deep learning approaches using public tweets to understand COVID-19 vaccine hesitancy.

### 2.2. Existing Gap in Literature

Considering previous research on COVID-19 vaccine hesitancy using a deep learning approach, as shown in Table 1, there seems to be a paucity in the literature that needs to be addressed:I.Some previous research projects were limited to only sentiment analysis. We believe that extracting emotions, in addition to its sentiments, will enhance the study.II.Some previous works were based on datasets downloaded from other institutions. We believe there is a need for a shift in the dataset used to analyze the study with updated records. Consequently, addressing updated COVID-19 vaccination hesitancy tweets is another crucial matter to be considered.III.Building complex emotions could enhance the discussions on COVID-19 vaccine hesitancy. Most previous studies were limited in scope on classifications. We believe that integrating the topic with another field, such as psychology, could enhance the research study.

From the gaps in the previous studies on COVID-19 vaccine hesitancy, as represented in Table 1, this study utilized text-based sentiment analysis and deep learning techniques to investigate COVID-19 public tweets. Our goal is to enhance our understanding of public opinions on COVID-19 vaccine hesitancy. Therefore, we designed, developed, and evaluated various models to ascertain the best deep learning model for the COVID-19 public tweets dataset.

## 3. Methodology

### 3.1. Overview

The experimental design of our project is shown in Figure 1. The first step, as shown in the figure, was text preprocessing. In this process, the dataset is cleaned by removing the URLs, punctuations, and retweets. In the preprocessing stage, the emojis in the dataset were converted to words. We also performed tokenization and removed stop words before performing stemming and lemmatization. Removing collection words was performed as well. After the text preprocessing, we classified the tweets using NRCLexicon into two sentiments and eight emotions. We computed word frequency for data exploration. Finally, the last step was to build neural networks using 1DCNN, LSTM, Multiple-Layer Perceptron (MLP), and BERT.

### 3.2. Data Collection

The dataset was collected from Twitter using API search words #vaccine or #COVID-19. We collected English public tweets globally. One of the limitations is that API search can only be retrieved in about seven to nine days, so we could only collect public tweets for up to nine days. We collected the dataset daily from 26 September 2021 to 27 March 2022. Using Twitter API search during this time frame, we collected one hundred fifty-six thousand five hundred thirty-nine public tweets and saved them as CSV (Comma-Separated Values).

Figure 2 represents the number of tweets collected on a daily basis. We notice that at the beginning of February 2022, the number of tweets decreases tremendously. We collected an average of one thousand tweets every day prior February 2022. Figure 2 shows that there is a decline in the number of tweets related to COVID-19 from February 2022 onward. There is a possibility that people started discussing other topics on Twitter rather than COVID-19. In the U.S., for example, people could be discussing tax return until mid-April. In addition, the end of February 2022 was the beginning of the Ukraine war. There is a possibility that people around the globe shifted the discussion from COVID-19 to topics related to the Ukraine War topic. Furthermore, it had been two years that we were dealing with COVID-19. The is a likelihood that people started feeling happy that COVID-19 cases had been decreasing, so the COVID-19 topic was no longer a critical issue.

### 3.3. Text Preprocessing

Text preprocessing is essential in Natural Language Processing (NLP) before performing text sentiment classification. In this step, texts are prepared to be trainable for machine learning algorithms.

The first step is removing retweets. A retweet is a tweet shared from one user to another. This will result in tweet duplications. Removing retweets will normalize the dataset. A dataset with normal distribution will improve the fitness of the model used. Another benefit of removing retweets is to minimize the space required to run the experiment. Consequently, we removed retweets to improve the classification models.

The second step is to remove URLs and punctuation. It is important to clean the URLs since each tweet has a link. Keeping links in the dataset will skew word frequency. Furthermore, removing non-useful signs and punctuation will eliminate unnecessary items in the dataset. Because URLs and punctuations have no meaning, removing unnecessary items may improve model efficacy.

The next step is converting emojis into words. We used the library emoji.demojize() in sklearn to convert emojis into words. This is necessary because some people tend to express their feelings using emojis. Transforming emojis into words might improve tweets’ sentiment analysis.

Next, we performed tokenization and normalization. Tokenization is a process of splitting words in each tweet, so it will be helpful to perform text-based sentiment analysis. Normalization is a process to normalize the collection of words by lowering the case as normalized words. Each word from the tweets is stored in a collection of words before computing its polarity.

After normalizing phrases, we removed stop words from the collection of words. Removing English stop words, such as ‘am’, ‘is’, ‘aren’t’, ‘hasn’t’, etc. will be helpful in computing text-based sentiments since these stop words are not valuable in this analysis. In addition, having stop words in the word collection will skew the frequency of the words. Consequently, removing them might improve text-based sentiment analysis.

The next step is stemming and lemmatization. Stemming is a process of transforming words into their base or root form. Words such as learning, learns, or learned are transformed to learn. Reducing the form of words to their base forms will homogenize the structure of the collection of words. We used PorterStemmer in this experiment. Unlike stemming, lemmatization is a process of reducing the form of words into the root of their dictionary forms. The purpose of performing lemmatization is to remove inflectional endings and return them to their dictionary base forms. WordNetLemmatizer was used for the experiment. Studies have shown that this process could enhance text-based sentiment analysis [6].

The final step of text preprocessing is to remove words used to search in the tweet collection. Keeping a collection of words, such as COVID-19, vaccine, and corona, will increase the skewness of word frequency. Consequently, removing these words will help normalize the distribution of the dataset.

### 3.4. NRCLexicon

The NRC Emotion Lexicon is a directory of English phrases and their relations with eight basic emotions and two sentiments (positive and negative) [56]. A lexicon is an emotional analysis that contains a set of words that will classify texts according to Plutchik’s Wheel of Emotion [57]. These eight basic emotions are joy, trust, fear, surprise, anticipation, anger, disgust, and sadness, which are popularly called Plutchik’s Wheel of Emotions [58]. The NRCLexicon technique classifies tweets associated with the NRC Emotion Lexicon directory into ten classes: positive sentiment, negative sentiment, and eight basic emotions (joy, trust, fear, surprise, anticipation, anger, disgust, and sadness) [59]. The 8 basic emotions resulting from the RNCLexicon are based on Plutchik’s research and his Wheel of Emotions. These eight primary emotions are grouped into opposites: joy–sadness, acceptance (trust)–disgust, fear–anger, and surprise–anticipation.

#### 3.4.1. Daily Emotions and Sentiments

Figure 3 shows the basic emotional effects from COVID-19 and its vaccines. It appears that the highest frequency is trust, followed by fear. Despite many conspiracy theories and myths, there is an indication that people have more trust that the pandemic and its vaccines are real (not a hoax). On the other hand, the lowest frequency is anger, followed by disgust. Even though individuals are dealing with hardship, people can hold their anger to the situation. Feeling angry arises when people are mistreated. Because most individuals trust the issue, it is most likely that people can hold their anger when coping with hardships. People tend to be more disgusted than angry when dealing with the COVID-19 pandemic and its vaccines. If there is an option, people do not want to get COVID-19 vaccines. However, it seems individuals in societies can hold their anger because they realize the benefits of getting vaccinated to combat the pandemic.

According to Plutchik’s Wheel of Emotions, the eight basic emotions are in opposite pairs (fear–anger, joy–sadness, trust–disgust, and surprise–anticipation). Figure 4 shows the emotional impact of COVID-19. The graph shows that fear and anger are expressed in about the same frequency. There is a possibility that people feel fearful and angry at the situation. People might fear the novel virus because its medications have not been found yet. At the same time, communities might be angry with the situation because there was not much they could do to combat the disease. There is a possibility that individuals are slightly more fearful when dealing with the pandemic and getting COVID-19 vaccines than being angry with the situation.

The second group of emotions (joy and sadness) in Figure 5 shows that people tend to be sad when dealing with the pandemic and its vaccines than being joyful. There is a likelihood that vaccine conspiracy theories, myths, and other types of anti-vaccination campaigns negatively impact societies. In addition, the use of vaccines as an emergency medication to combat COVID-19 is still being debated, where some groups of people still resist getting vaccinated. It makes sense that people are more heartbreaking when dealing with a new disease than being joyful.

The next emotional impact is trust and disgust. According to Figure 6, there is a slight possibility that people tend to trust or be more accepting than being disgusted when dealing with the COVID-19 pandemic and its vaccines. Individuals in communities accept (trust) the reality that COVID-19 does exist, and not fake news (hoax). People tend to act based on what they believe, so accepting the reality that the pandemic is real might influence more people to accept COVID-19 vaccines as an emergency treatment to combat the pandemic.

The last emotional impact of COVID-19 is surprise–anticipation. Figure 7 shows that anticipation is more frequent than surprise. There is a probability that communities have been more anticipative (being predictable) than surprised by the news of the pandemic and its vaccines. Because the COVID-19 pandemic is something new to society, there is a possibility that people are more anticipated that there could be an emergency treatment to combat the disease. Individuals in societies are not surprised by any new medication or invention to quickly fight the COVID-19 pandemic.

Figure 8 (daily positive–negative sentiments) shows that positive daily sentiments are more frequent than negative sentiments. The graph demonstrates that the number of positive tweets is consistently more than the number of negative sentiments. It indicates that people feel more positive about getting COVID-19 vaccines as an emergency prevention of getting the novel virus. Even though people are dealing with such hardship during the pandemic, people feel more optimistic than thinking negatively about the COVID-19 issue. Because people trust that the pandemic is real, not a hoax, communities feel optimistic that they can cope with the difficulties.

To test the pairs of daily emotions, we performed a *t*-test to check if the relations among the basic emotions are statistically significant or not. Table 2 shows that the *p*-values of the relations among the pairs of emotions resulting from COVID-19 are significant. According to the *t*-test results, we can confirm that people have been sad and more miserable than joyful when dealing with pandemic. Facing the COVID-19 pandemic, people have more trust that COVID-19 is a real issue than disgust that the novel virus is a conspiracy. Furthermore, individuals in societies have been more worried (fearful) when coping with the pandemic than angry with the situation. In addition, because more people trust that the novel virus is a real issue, the COVID-19 pandemic has been anticipated rather than being a surprising topic. Finally, the *p*-value of the negative–positive relation is also close to zero. It suggests that there is significant evidence that people feel optimistic about getting the COVID-19 vaccines, and they trust these vaccines as an emergency treatment to prevent the spread of the novel virus.

#### 3.4.2. Complex Emotional Effects

Complex emotion [60] is the combination of two basic emotions (anger, fear, trust, disgust, surprise, anticipation, joy, and sadness) blending into a new formation of emotions. In addition, the American Psychological Association [60] suggests that the evolutions of new compositional emotions are seen more often in older adults than children. Aggressiveness is a mixture of anticipation and anger, while contempt is a fusion of disgust and anger. In addition, remorse is a blend of sadness and disgust, while disapproval is a mix of surprise and sadness. Next, awe is a formulation of fear and surprise, while submission is a mixture of trust and fear. The combination of joy and trust forms love, while the unification of joy and anticipation forms optimism.

Figure 9 shows the complex emotions resulting from the tweets. The graph indicates that most people feel submissive to COVID-19 and its vaccines. Society members trust that the pandemic is real, but they fear the situation. Consequently, most people feel obedient to the rules and regulations related to COVID-19. The next larger area in the graph is love. Because people feel trust about an emergency solution to combat the spread of the novel virus and are slightly joyful with the benefits of the vaccines, many individuals in communities love to have COVID-19 vaccines. People feel more submissive than loving because they have more fear than joy. The smallest area of the complex emotional effects is contempt. According to the complex emotions, contempt has the lowest area because anger is the lowest emotion, and disgust is the second-lowest one. It indicates that people tend to be contempt of the pandemic and feel optimistic about dealing with COVID-19, especially with the benefits of getting COVID-19 vaccines.

### 3.5. Neural Network Models

The next stage in this project was building neural network modeling. We built three deep learning approaches: LSTM (Long Short-Term Memory), Multiple-Layer Perceptron, and BERT (Bidirectional Encoder Representations from Transformers). We tested the three methods and compared which method had the best performance.

#### 3.5.1. 1DCNN

In this neural network model, we embedded a maximum of twenty thousand words and applied a 1D convolutional layer with a kernel size of 1 and ReLu activation function. In addition, 1D MaxPooling used a pool size of 1. Next, we applied a dense layer with a unit size of thirteen, and the activation functions were ReLu. Lastly, we used another dense layer with a unit size of ten and applied sigmoid activation. Figure 10 shows the architecture of the proposed 1DCNN model.

#### 3.5.2. LSTM (Long Short-Term Memory)

LSTM is a special type of advanced Recurrent Neural Network (RNN), a sequential network, that can handle long-term dependencies, which comprise the vanishing gradient encountered by an RNN. Similarly, an LSTM model remembers the previous information and uses it to process the current input, but it is designed to avoid the long-term dependency issue that is RNN’s weakness. In this neural network model, we embedded a maximum of fifty thousand words with an embedding dimension of one hundred words, and a maximum sequence length of two hundred and fifty words. In addition, we applied a 1D spatial dropout of 0.2 before the LSTM layer. We applied one-hundred-unit size in the LSTM layer, with dropout of and recurrent dropout of 0.1. Before the last layer, we add another dropout of 0.3. Finally, we applied a dense layer with a size of ten and SoftMax activation function. Figure 11 shows the architecture of the proposed LSTM model.

#### 3.5.3. Multiple-Layer Perceptron

A Multiple-Layer Perceptron (MLP) in a neural network is a layer deeply connected with its previous layers. It means that the neurons of the layers are connected to each neuron of its preceding layer. Matrix-vector multiplication is applied where the row vector of the output from previous layers is equal to the column vector of the MLP. Figure 12 and Figure 13 show the MLP model.

We started with two layers using a sequential model in this neural network. The unit size in the first layer was sixteen with a ReLu activation function. Before the second layer, we added a dropout of 0.43. Lastly, the final layer with the size of ten was activated with SoftMax. Secondly, we applied three MLP layers of neural networks. We started with a unit size of 128 with a ReLu activation function. Next, we applied a dropout of 0.61 before adding another MLP layer. The second MLP layer size was thirteen with a ReLu activation function. Before the last layer, we added a dropout of 0.1. Finally, we added the last layer with the size of ten and an activation function of SoftMax.

#### 3.5.4. BERT (Bidirectional Encoder Representations from Transformers)

BERT (Bidirectional Encoder Representations from Transformers) is a published project by the authors of Google AI Language. The primary key to this innovation is applying the bidirectional training of Transformers (a trendy attention model). In this neural network model, we applied seven hundred and sixty-eight sizes of input, followed by a hidden layer with fifty units and an output layer. In this neural network model, we applied an Adam optimizer with a learning rate of 3 × 10^−5^ and decay of 1 × 10^−8^.

## 4. Results

In this section, we show the results of the experiments. We examined the result by applying neural network experimentations, including 1DCNN, MLP, LSTM, and BERT, using ten classes (positive, negative, joy, sadness, trust, disgust, fear, anger, surprise, and anticipation).

This section discusses our neural network experimentations on COVID-19 tweets. We examined the performances of the 1DCNN, MLP, LSTM, and BERT models. The first experiment was when we applied the 1DCNN model. According to this experiment, we achieved the highest accuracy of 0.886133. Additionally, we achieved the highest performance with a training time of 1744 s. Figure 14 and Figure 15 show the 1DCNN accuracy and loss graphs.

Our next model performance was the LSTM model. Our experimentation on this deep learning model achieved the highest accuracy of 0.8993. To accomplish this performance, it took 27,597 s for five epochs. From Figure 16 and Figure 17, we notice that the model converges when the training is about 3.7 epochs.

The subsequent model experimentation was the MLP model. In this experiment, we applied two and three dense layers of artificial networks. When we applied the two layers of the MLP model, we achieved the highest accuracy of 0.8478 with a training time of 203 s. On the other hand, when we applied the MLP model with three layers, we accomplished an accuracy of 0.8117 with a training time of 1405 s. When we applied two MLP layers, the model accomplished faster than the model with three dense layers because the data had to go through another model layer. Figure 18 and Figure 19 show the MLP accuracy and loss graphs with two dense layers, while Figure 20 and Figure 21 represent MLP accuracy and loss graphs of the three dense layers.

Finally, the last model experimentation was the BERT model (Figure 22). In this neural network model, we accomplished an accuracy of 0.9671 with a training time of 8429 s. Applying this model, we achieved the highest model performance overall. The training time was slightly longer than some other models, but the LSTM model had the longest training time (27,597 s), which was almost three times the BERT training period.

## 5. Discussions

For our first model experiment with 1DCNN, the result of this neural network model is higher than some previous deep learning models, with an area under the curve of 0.7600 for three classes (positive, negative, and neutral) [54]. In this experiment, we used ten classes (two sentiment and eight emotions), but our model was able to perform higher than some previous research studies.

In addition, for our second model experiment with LSTM, the model’s performance is better than some previous studies: 0.7544 [49], 0.6500 [48], 0.8660 [54], and 0.7752 [43]. One model [55] has an LSTM accuracy of 0.9167. However, other studies only used two classes (positive and negative), while our study applied ten classes (two sentiments and eight emotions).

Our experiment showed that two layers of MLP achieved the fastest time (203 s) to reach the maximum result of 0.8478, while the three layers of MLP achieved its best performance (accuracy of 0.8117) within 1405 s. The other models took a long time to achieve their best model performances, such as the 1DCNN model took 1744 s, the BERT model took 8429 s, and the LSTM model took 27,597 s. In terms of time efficiency, the MLP model with two layers performed the most efficiently, and the LSTM model was the most inefficient or time-consuming.

According to our results, BERT was the best model classifier in our experiment. The BERT model’s achievement is better than most of the previous BERT model accuracies: 0.9235 [45], 0.9160 [42], 0.9150 [52], and 0.8900 [44]. One BERT model [47] achieved a similar accuracy (97%) as our model, but this previous experiment used two classes only (positive and negative) when compared with our experiment (two sentiments and eight emotions).

On the other hand, the MLP model with three layers achieved the lowest model performance (0.8117). The second highest model performance was achieved using LSTM (0.8993), followed by 1DCNN (0.886133). This model’s achievement is like a previous research experiment [54], but the previous study only applied two classes (positive and negative), while our experiment used ten classes (two sentiments and eight emotions).

Comparing our experimentation on several neural networks, the BERT model accomplished the highest model accuracy of 0.9671 with ten classes (two sentiments and eight emotions). Our next highest model accuracy was achieved by the LSTM model (0.8993). However, the LSTM model’s training time was time-consuming as it was three times longer than the BERT model’s training period. We consider the BERT model as the best model experimentation based on our dataset, even with a relatively long training time.

## 6. Contributions

According to our experimental results, below are some of the contributions of this study:We designed, developed, and evaluated deep neural network models with ten classes (positive, negative, joy, sadness, trust, disgust, fear, anger, surprise, and anticipation). Most previous studies were limited to three classes of sentiment (positive, negative, and neutral). For instance, [38] analyzed a Twitter dataset based on VADER sentiment computation with three classes (positive, negative, and neutral) using LSTM and BiLSTM neural networks. Ref. [54] performed sentiment analysis using three sentiment classes (positive, negative, and neutral) by applying a deep neural network model. Ref. [52] applied ANN, LSTM, and BERT neural networks using four classes (pain, fever, fatigue, and headache). Ref. [44] experimented with BERT, LinearSVM, Logistic Regression, and LSTM using four classes (sad, joy, fear, and anger). Ref. [39] classified their research’s sentiments into five classes (positive, extremely positive, negative, extremely negative, and neutral).The proposed deep neural network models demonstrated an improvement over previous research studies. For example, [41] performed modeling, such as Random Forest, XGBoost, LinearSVC, Decision Tree, LSTM, BiLSTM, and CNN-LSTM, with three classes (positive, negative, and neutral) using TextBlob sentiment computation. The best accuracy of their work was 93%. Ref. [42] performed LinearSVC, Multinomial Naïve Bayes, BiLSTM, and BERT models with three classes of sentiments using TextBlob computation. Their best performance was the BERT neural network with an accuracy of 91.6%. Furthermore, [45] used Decision Tree, GaussianN, Random Forest, KNN, Fusion Model, Simple Neural Network, CNN, LSTM, and BERT. The best performance was achieved by the BERT neural network with an accuracy of 92.35%. Our research study performed better (accuracy of 96.71%) when using the BERT neural network model.Our research study identified eight basic emotions from the tweets (joy, sadness, trust, disgust, fear, anger, surprise, and anticipation) in addition to the two sentiments (positive and negative). Furthermore, we extended the basic emotions into complex ones (aggressiveness, contempt, remorse, disapproval, awe, submission, love, and optimism). Most of the previous classes were based on the sentiments only (positive, negative, and neutral) [45,51,54,55]. Another study classified their dataset into four basic emotions (sad, joy, fear, and anger) [44]. Finally, a study used a combination of primary and complex emotions (optimistic, thankful, emphatic, pessimistic, anxious, sad, annoyed, denial, official, report, surprise, and joking) [40]. Our research is based on a ten-class solution using deep learning models of LSTM, BiLSTM, and BERT to classify COVID-19 vaccine hesitancy tweets into positive, negative, joy, sadness, trust, disgust, fear, anger, surprise, and anticipation.

## 7. Conclusions

The COVID-19 pandemic has impacted the global community tremendously. Various rules and regulations forced many business entities to change their strategies to survive while coping with the disruptions and hardships impacted by the coronavirus disease. Conspiracy theories, anti-vaccination campaigns, myths, or controversies regarding the safety issue of COVID-19 vaccines made some people hesitate to get vaccinated. Other groups supporting vaccination kept encouraging the use of vaccine to combat the coronavirus disease. Examining the level of COVID-19 vaccine hesitancy is an effort to combat this infectious virus. Since there are pros and cons of getting COVID-19 vaccines, examining vaccine hesitancy using social media dataset becomes an essential topic.

Our study indicates that people gradually feel positive about COVID-19 vaccine, shifting their hesitancy concerns and getting COVID-19 vaccines due to improved public trust in the benefits of getting vaccinated. Most people trust that the pandemic is a real issue, and not a hoax. Individuals in societies can hold their anger when dealing with COVID-19 vaccines. In addition, most people feel submissive when dealing with COVID-19 and its vaccines. Second, the majority of people feel loving when coping with the novel virus and its vaccines. Few people feel contempt because they are able to hold their anger when facing COVID-19 and its vaccinations. Daily tweet sentiment analysis shows significant evidence that communities have been optimistic about getting vaccinated as an emergency exit to combat COVID-19.

At the beginning of February 2022, there was an indication that people reduced their discussion about COVID-19 on social media platforms, such as Twitter. This could be attributed to various factors, such as people shifting their opinions as the pandemic felt less scary than before at the beginning of 2019. In addition, discussions on social media shifted to new hot topics, such as the war between Ukraine and Russia, after lingering and chatting about coronavirus for a couple of years.

Based on our experimental results, we successfully attained a considerably better performance in our sentiment analysis on COVID-19 vaccine hesitancy. Our findings show that compared to supervised learning, artificial neural network techniques have better computational performance, as demonstrated in the present study using sentiment analysis and model validation. Our 1DCNN model showed higher performance when compared to other deep learning models in related works. The application of LSTM also achieved a more comprehensive sentiment analysis of 10 classes of emotions, giving a broader outlook of classified sentiments, when compared to studies that had binary classification (positive and negative). Finally, our neural network experimentation showed that BERT performed better than the other neural network models. The BERT model with ten classes achieved the highest accuracies of 0.9671 which is higher than previous research studies (mostly using three classes). The training time to achieve its highest performance was 8429 s. Our second highest accuracy was achieved by the LSTM model (0.8993), which had an extremely long training time of 27,597 s.

## 8. Future Work

To improve the current research study, we would like to implement a time series analysis on companies related to COVID-19 vaccine stock prices in future studies. We would like to gather various COVID-19 vaccine companies’ stock prices, and not only from the three major companies in the U.S. (Pfizer, Johnson and Johnson, and Moderna).

The worldwide COVID-19 pandemic has impacted the global community tremendously. Schools and business closures have caused a significant transformation in societies. Various rules and regulations have forced many business entities to change their strategies to survive while coping with the pandemic.

## Figures and Tables

**Figure 1 ijerph-20-05803-f001:**
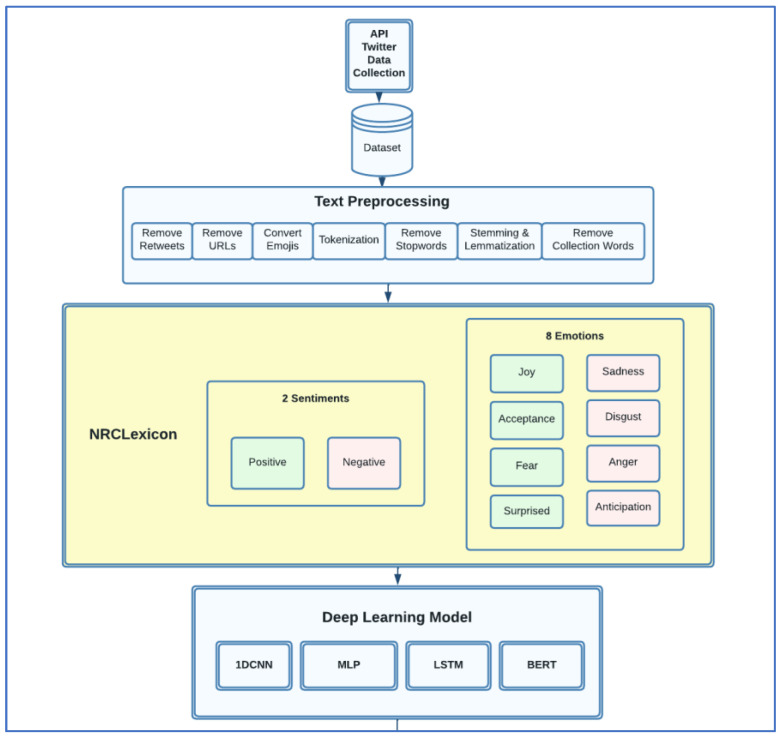
Experimental Design.

**Figure 2 ijerph-20-05803-f002:**
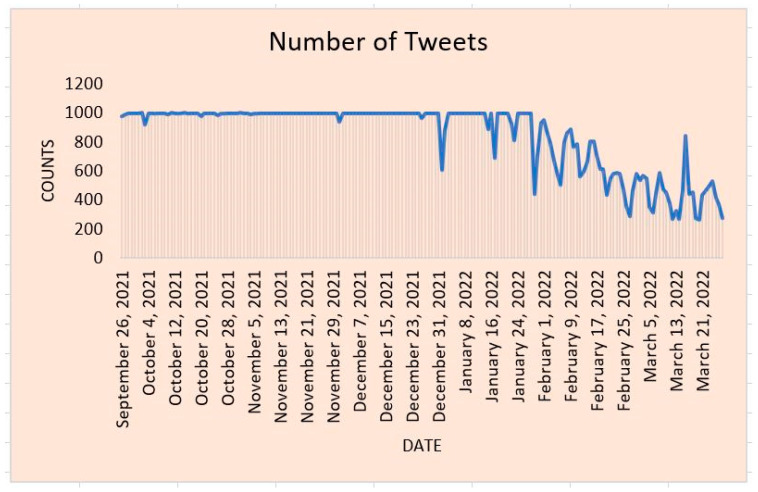
Number of COVID-19 Tweets.

**Figure 3 ijerph-20-05803-f003:**
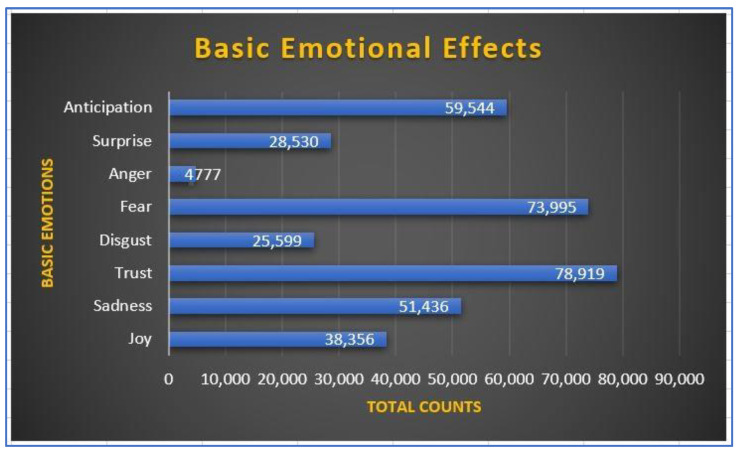
Basic Emotional Effects.

**Figure 4 ijerph-20-05803-f004:**
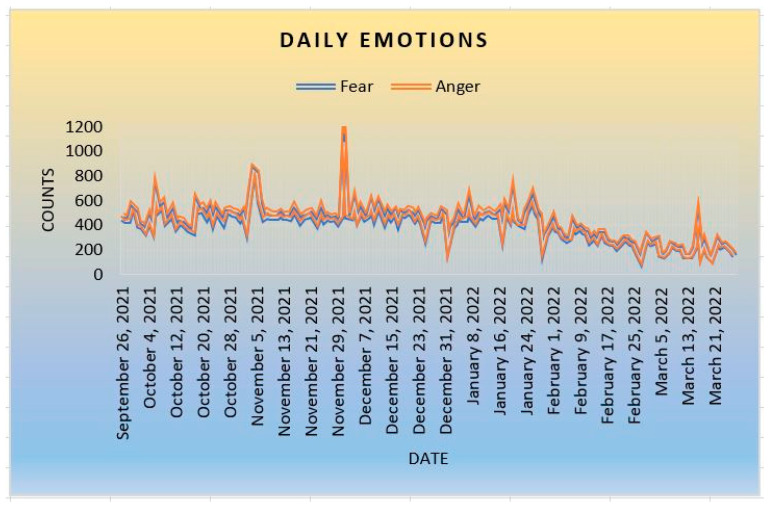
Fear and Anger.

**Figure 5 ijerph-20-05803-f005:**
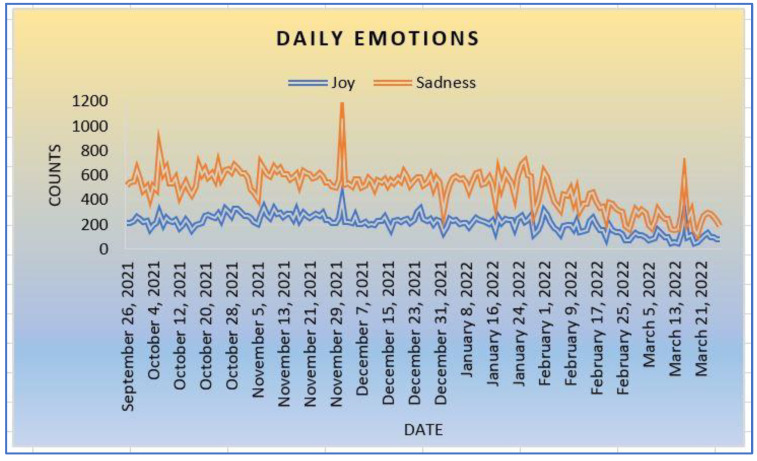
Joy and Sadness.

**Figure 6 ijerph-20-05803-f006:**
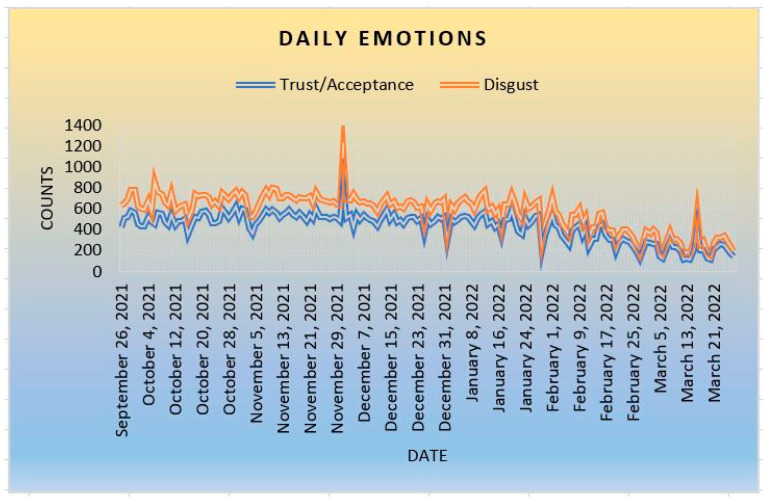
Trust and Disgust.

**Figure 7 ijerph-20-05803-f007:**
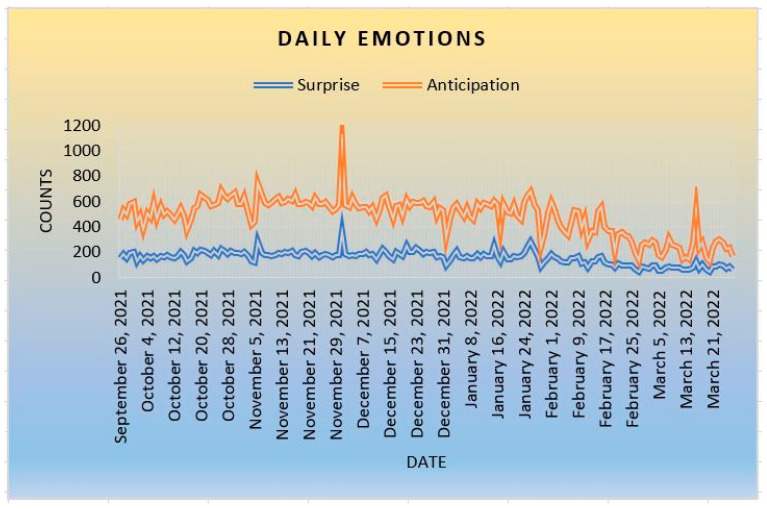
Surprise and Anticipation.

**Figure 8 ijerph-20-05803-f008:**
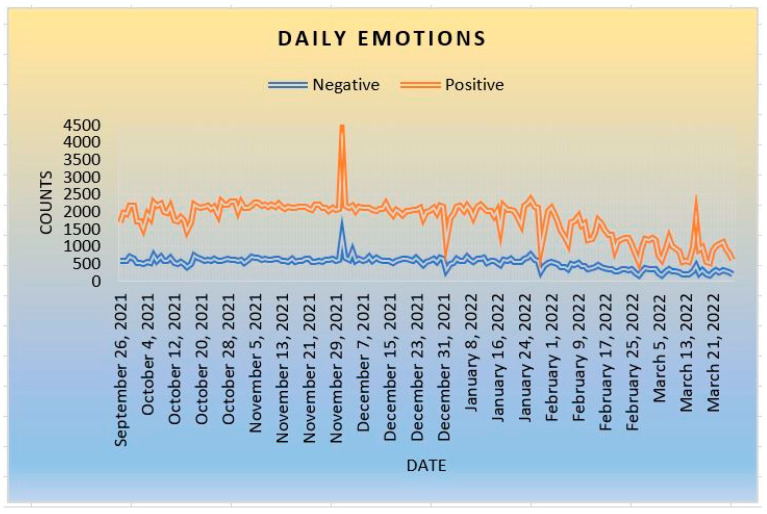
Daily Sentiments.

**Figure 9 ijerph-20-05803-f009:**
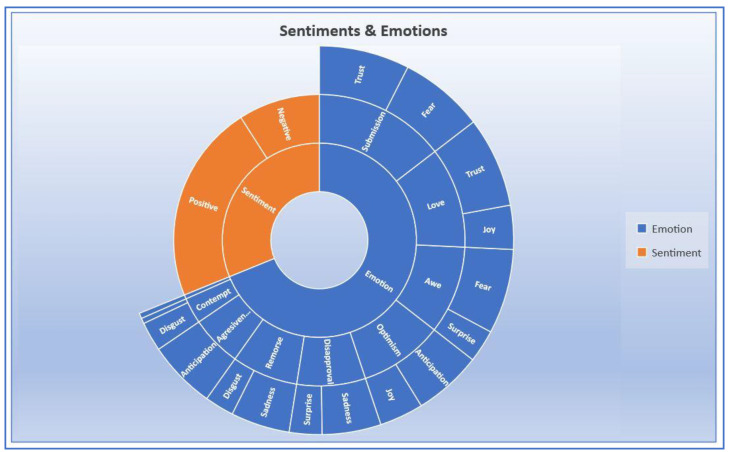
Complex Emotions.

**Figure 10 ijerph-20-05803-f010:**
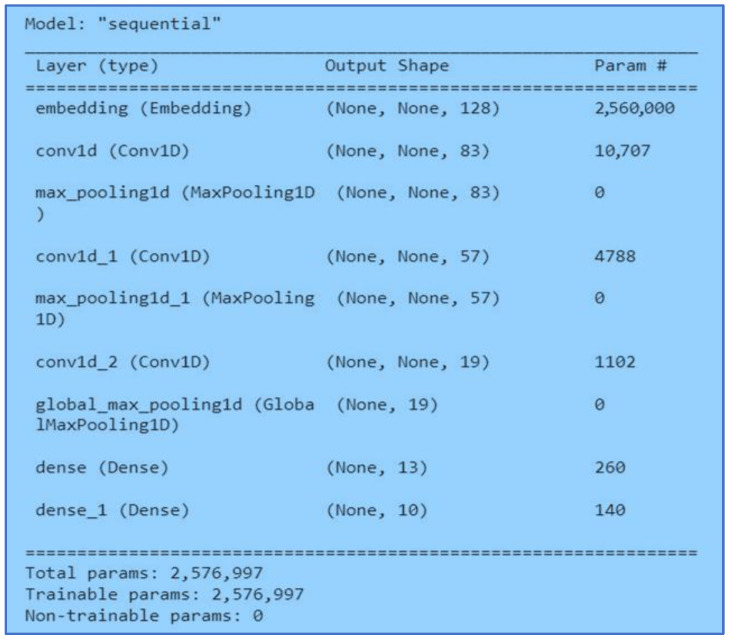
1DCNN Model.

**Figure 11 ijerph-20-05803-f011:**
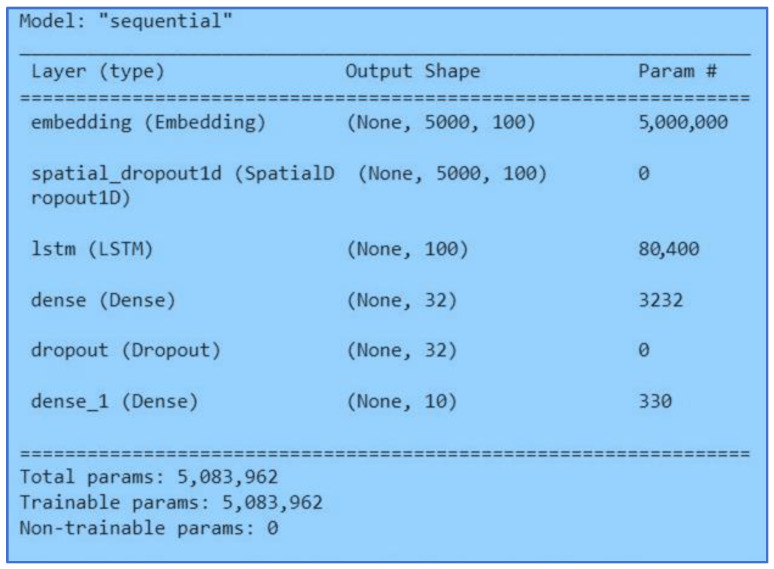
LSTM Model.

**Figure 12 ijerph-20-05803-f012:**
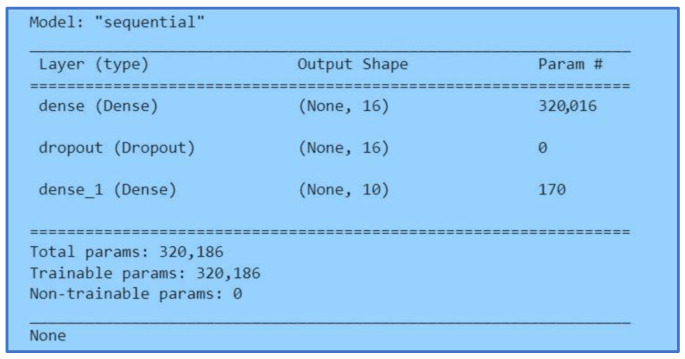
MLP Two Layers Model.

**Figure 13 ijerph-20-05803-f013:**
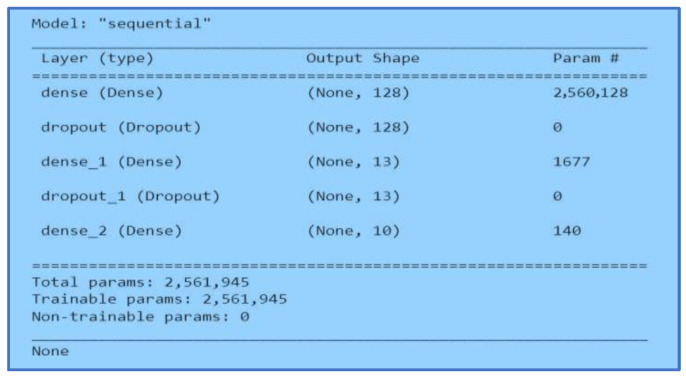
MLP Three Layers Model.

**Figure 14 ijerph-20-05803-f014:**
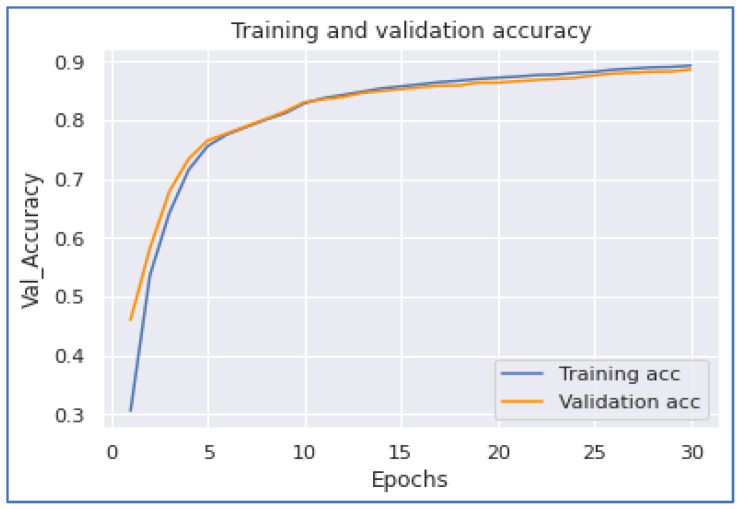
1DCNN Accuracy.

**Figure 15 ijerph-20-05803-f015:**
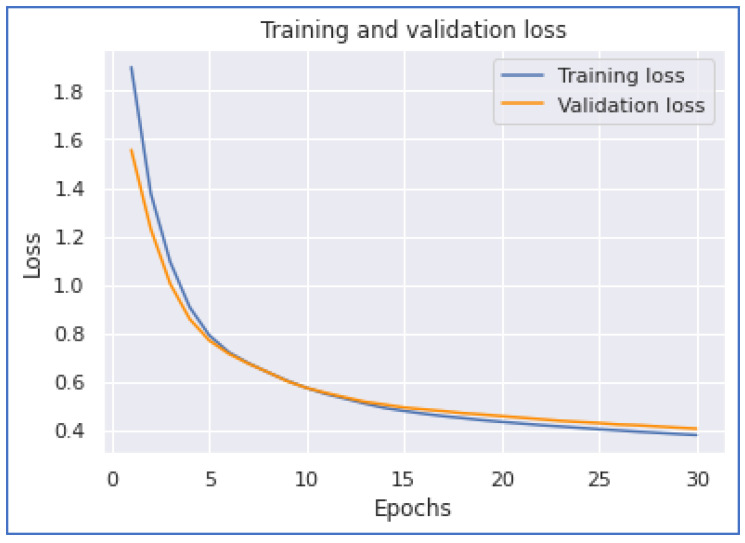
1DCNN Loss.

**Figure 16 ijerph-20-05803-f016:**
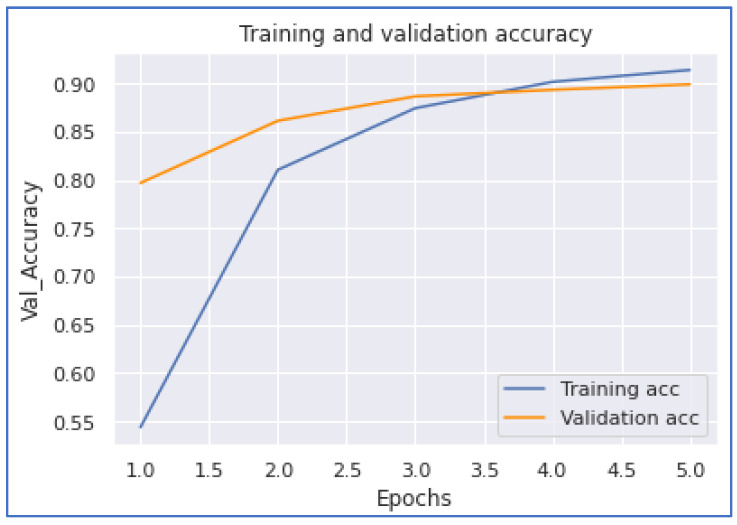
LSTM Accuracy.

**Figure 17 ijerph-20-05803-f017:**
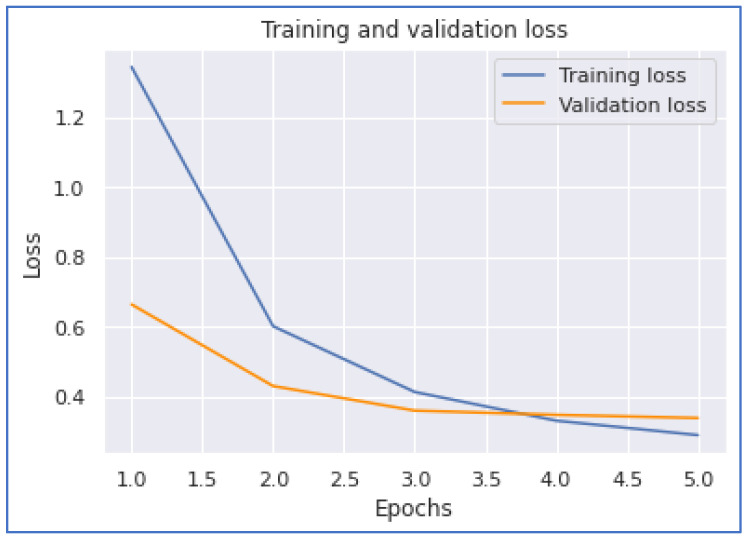
LSTM Loss.

**Figure 18 ijerph-20-05803-f018:**
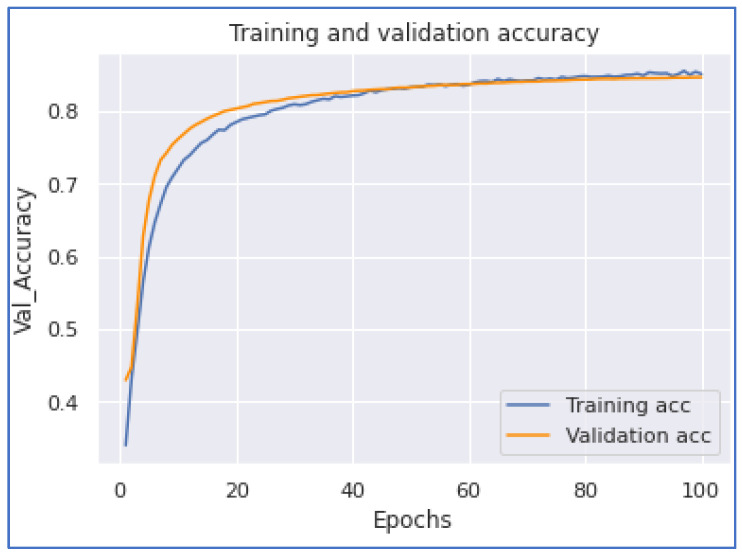
MLP Accuracy (Two Dense).

**Figure 19 ijerph-20-05803-f019:**
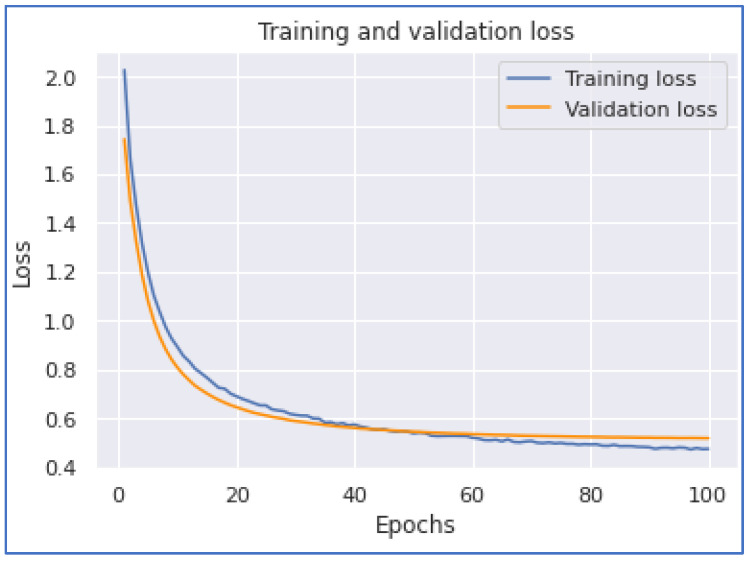
MLP Loss (Two Dense).

**Figure 20 ijerph-20-05803-f020:**
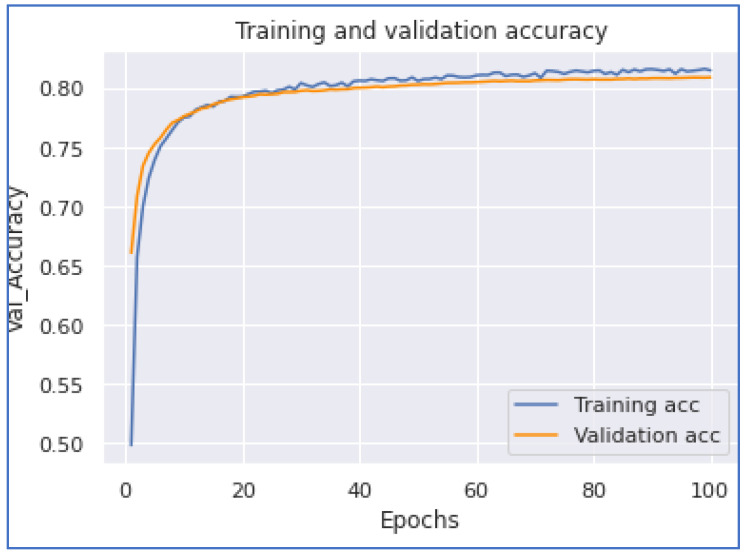
MLP Accuracy (Three Dense).

**Figure 21 ijerph-20-05803-f021:**
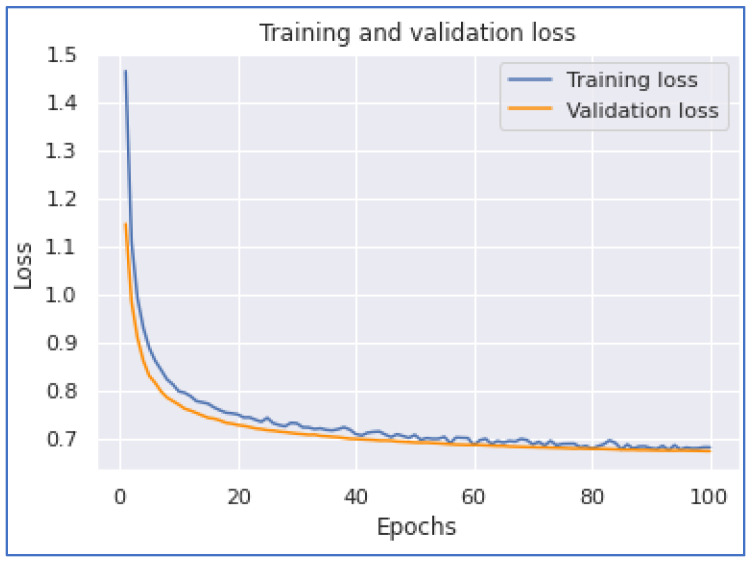
MLP Loss (Three Dense).

**Figure 22 ijerph-20-05803-f022:**
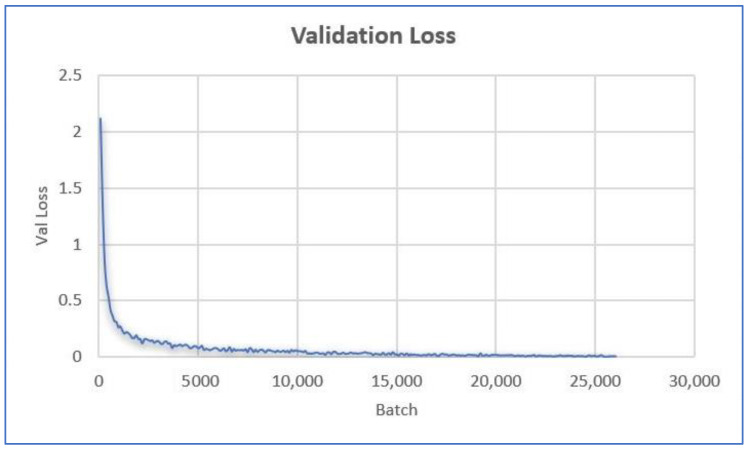
BERT Validation Loss by Batch.

**Table 1 ijerph-20-05803-t001:** Deep Learning Studies Using COVID-19 Sentiment Analysis.

No.	Title	Authors	Datasets	Classes	Method	Conclusion
1	A performance Comparison of Supervised Machine Learning models for COVID-19 Tweets Sentiment Analysis	Furqan Rustam, Madiha Khalid, Waqar Aslam, Vaibhav Rupapara, Arif Mehmood, and Gyu Sang Choi.	IEEE data port, 7528 tweets	Positive, negative, and neutral	RF, XGBoost, LinearSVC, ETC, DT, LSTM, BiLSTM, and CNN-LSTM	ETC achieves the highest accuracy of 93%, LSTM accuracy is 57.7%, BiLSTM accuracy is 57.9%, and CNN-LSTM accuracy is 61% [41].
2	Applying Machine Learning to Identify Anti-Vaccination Tweets during the COVID-19 Pandemic	Quyen G. To, Kien G. To, Van-Anh N. Huynh, Nhung T. Q. Nguyen, Diep T. N. Ngo, Stephanie J. Alley, Anh N. Q. Tran, Anh N. P. Tran, Ngan T. T. Pham, Thanh X. Bui, and Corneel Vandelanotte.	1,651,687 English tweets	Positive, negative, and neutral	SVC, NB, BiLSTM, and BERT	BERT has the highest accuracy performance of 91.6% [42].
3	A Hybrid Feature Extraction Method for Nepali COVID-19-Related Tweets Classification	Tej Bahadur Shahi, Chiranjibi Sitaula, and Nawaraj Paudel.	Kaggle, 33,458 tweets	Positive, negative, and neutral	Regression, RF, KNN, Naïve Bayes, SVM AdaBoost, and MLP-ANN	SVM + RBF model has the highest precision of 74.4%, recall of 76.9%, and F1-score of 75.6% [37].
4	Examining Rural and Urban Sentiment Difference in COVID-19—Related Topics on Twitter: Word Embedding—Based Retrospective Study	Yongtai Liu, Zhijun Yin, Congning Ni, Chao Yan, Zhiyu Wan, and Bradley Malin.	*Tweepy* Python library, 407 million Geotagged tweets	Positive and negative	Vector Clustering and Inference Analysis	The study showed that there was a statistically significant difference in the sentiments of urban and rural Twitter users regarding a wide range of COVID-19–related topics with *p*-value of *p* < 0.001 [46].
5	Deep Learning Model for COVID-19 Sentiment Analysis on Twitter	Salvador Contreras Hernández, María Patricia Tzili Cruz, José Martín Espínola Sánchez, and Angélica Pérez Tzili.	33,776 tweets	Positive and negative	SVM + Naïve Bayes, Logistic Regression, Decision Trees, and BERT	BERT model achieves 97% accuracy in training and 81% in testing [47].
6	Deep Learning-Based Sentiment Analysis of COVID-19 Vaccination Responses from Twitter Data	Kazi Nabiul Alam, Shakib Khan, Abdur Rab Dhruba, Mohammad Monirujjaman Khan, Jehad F. Al-Amri, Mehedi Masud, and Majdi Rawashdeh.	Kaggle, 125,906 tweets	Positive, negative, and neutral	LSTM and BiLSTM	The accuracy of LSTM is 90.59%, and BiLSTM has an accuracy of 90.83% [38].
7	COVID 19 Tweets Classification Using RNN in Deep Learning	S. Kiruthika Devi, Aditya Upadhyay, and Saket Dimri.	288,500 tweets	Positive, extremely positive, negative, extremely negative, and neutral	LSTM	The model has a precision of 77.52% and an F1-score of 77% [39].
8	COVID-19 sentiment analysis via deep learning during the rise of novel cases	Rohitash Chandra and Aswin Krishna.	Lamsal R datasets consisting of 150,000 tweets from India, 18,000 tweets from Maharashtra (state), and 18,000 tweets from Delhi	Optimistic, thankful, emphatic, pessimistic, anxious, sad, annoyed, denial, official report, surprise, and joking	LSTM, BiLSTM, and BERT	The most tweets are associated with joking, optimistic, or annoyed, and a few are associated with thankful [40].
9	COVID-19 Vaccine Hesitancy in the Month Following the Start of the Vaccination Process	Liviu-Adrian Cotfas, Camelia Delcea, and Rares Gherai.	1,221,694 cleaned tweets	In favor, neutral, and against	MNB, RF, SVM, BERT and RoBERTa	The model that has the highest accuracy is RoBERTa (78.63%). Negative sentiments are associated with mistrust, freedom, side effects, hiding relevant information, unsafety, inefficiency, existence of alternatives, scam, and moral and religious issues [43].
10	Deep Learning-Based Methods for Sentiment Analysis on Nepali COVID-19-Related Tweets	C. Sitaula, A. Basnet, A. Mainali, and T. B. Shahi.	624,316 tweets	Positive, negative, and neutral	LinearSVM, RBF (Radial Basis Function) SVM, XGBoost, Artificial Neural Networks, RF, NB, LR, and KNN	RBF-SVM has the highest precision (70.2%), while the highest accuracy is XGBoost (66.7%). The accuracy of CNN-c is 68.7%, while CNN-ft has an accuracy of 68.1% [51].
11	Mild Adverse Events of Sputnik V Vaccine in Russia: Social Media Content Analysis of Telegram via Deep Learning	Andrzej Jarynowski, Alexander Semenov, Mikołaj Kamiński, and Vitaly Belik.	11,515 self-reported Sputnik on Telegram	Pain, fever, fatigue, and headache	ANN, LSTM, and BERT	Russian Telegram users reported mostly post-pain, fever, and fatigue after the Sputnik V vaccination. BERT has a precision of 91.5%, while LSTM has a precision of 86.6% [52].
12	Sentiment Analysis of COVID-19 Tweets Using Evolutionary Classification-Based LSTM Model	Arunava Kumar Chakraborty, Sourav Das, and Anup Kumar Kolya.	160,000 tweets	Positive and negative	LSTM	The model (LSTM) has an accuracy of 91.67% with validation accuracy of 84.46% [53].
13	Sentimental Analysis of COVID-19 Tweets Using Deep Learning Models	Nalini Chintalapudi, Gopi Battineni, and Francesco Amenta.	3090 tweets from GitHub	sad, joy, fear, and anger	BERT, SVM, LR, and LSTM	The highest model accuracy is BERT 89%. LR has an accuracy of 75%, SVM has an accuracy of 74.75%, and LSTM accuracy is 65% [44].
14	Sentiment Analysis of Bangladesh- specific COVID-19 Tweets Using Deep Neural Network	Muhammad Nazrul Islam, Ayon Roy, Saddam Hossain Mukta, Nafiz Imtiaz Khan, MD. Mahbubar Rahman, and A. K. M. Najmul Islam.	677 positive tweets, 921 negative tweets, and 256 neutral tweets	Positive, negative, and neutral	DNN	Area under the curve is 76%, with 55% people expressing negative sentiment on COVID-19, 38% expressing positive sentiment, and 7% expressing neutral sentiment [54].
15	Spatiotemporal Sentiment Variation Analysis of Geotagged COVID-19 Tweets from India Using a Hybrid Deep Learning Model	Vaibhav Kumar	128,096 tweets	Positive and negative	BiLSTM + CNN, LSTM, and CNN	The highest model accuracy is BiLSTM + CNN (89.68%) [55].
16	Text Classification Models for Automatic Detection of Fake COVID Products and News on Social Media	Kruthika Madhusudhana	Twitter API	Positive, negative, and neutral	DT, GaussianN, RF, K-Nearest, Fusion Model, Simple Neural Network, CNN, LSTM, and BERT	The model with the highest accuracy is BERT (92.35%). LSTM accuracy is 75.44%, Simple Neural Network has an accuracy of 65.17%, and CNN has an accuracy of 63.75% [45].

**Table 2 ijerph-20-05803-t002:** *p*-Values for Emotions and Sentiments.

Relations	Joy–Sadness	Trust–Disgust	Fear–Anger	Surprise–Anticipation	Negative–Positive
*p*-Value	6.42904 × 10^−17^	1.03093 × 10^−74^	5.73314 × 10^−89^	6.92704 × 10^−56^	1.07326 × 10^−69^

## Data Availability

The dataset used in this research has been uploaded on ResearchGate at https://www.researchgate.net/publication/364110620_COVID-19_Public_Tweets (accessed on 11 October 2022), DOI: 10.13140/RG.2.2.16612.86402.

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
