# Peer review of "COVID-19 Vaccine Hesitancy: A Global Public Health and Risk Modelling Framework Using an Environmental Deep Neural Network, Sentiment Classification with Text Mining and Emotional Reactions from COVID-19 Vaccination Tweets"

_ijerph, 2023, doi:10.3390/ijerph20105803_

Round 1
Reviewer 1 Report
COVID-19 Vaccine Hesitancy: A Deep Neural Network Sentiment Classification with Text Mining and Emotional Reactions from COVID-19 Vaccination Tweets
-----------------------
The paper proposes a deep learning model for sentiment classification on covid-19 tweet datasets. The paper is relevant to the current situation and has great value. However, it requires a major revision before acceptance.
1. It is unclear what is the open question in the abstract. There are several methods proposed in the literature, but why this method is required to deal with sentiment issue?
2. The contributions should be presented in bullet form in the introduction to make it clear and easy to understand. Please revise the introduction accordingly.
3. The small paragraphs in the introduction are like orphans, please merge them with a appropriate coherency in the introduction.
4. The paper requires to discuss the recent paper, A Hybrid Feature Extraction Method for Nepali COVID-19-Related Tweets Classification (hindawi.com)
to highlight issues raised by Covid-19 and state of the art models being used in the litearture.
5. As pointed previously, it is better to put the contributions in the introduction rather than as a separate section.
6. Please improve the graphics in the paper
7. The screenshot of the models in the paper are making the paper rough, please put high-quality models and figures instead of screenshots.
8. The paper requires more comparative study to show the validity of the findings.
Author Response
Good evening,
Herewith I attach the response. Thank you.

Reviewer 2 Report
Manuscript on an important topic, however, in its current form, the work is not suitable for publication. The following improvements are required.
The abstract needs some refinement. It is recommended that the authors clearly state what the purpose of the manuscript is, what methodology they used, and what results they achieved. Introduction. The authors do not specify what is novel in the work and what gap the work fills.
The authors in line 24 write that: "Neural network modeling, such as 1DCNN, LSTM, Multiple Layer Perceptron, and BERT, using ten classes (positive, negative, joy, sadness, trust, disgust, fear, anger, surprise, and anticipation) was applied in the research study"... the text lacks the development of the issues on the theoretical side.
Sentences line 28- 30: "There is significant evidence that people tend to express positive sentiments about the Covid-19 vaccine." People trust that Covid-19 is a real issue, not a hoax 29 and its vaccine is an effective emergency treatment to combat the pandemic' are unclear. It is not known whether these are the results of the authors' research or statements in relation to the literature. Test results should be clearly stated. If, however, the statements are based on literature, then these sentences should not be included in the abstract but rather in the introduction and supported by relevant literature.
There is significant evidence that people tend to express 28 positive sentiments about the Covid-19 vaccine.
Literature review. This section is definitely the weak side of the work. The authors refer to only a few references from 2022 and none from 2023. This section needs a lot of polishing.
Methodology is a strong point of the work. Presented in a legible, well-maintained, and attractive way. Figure 2. Number of Covid-19 Tweets - the x-axis with dates is illegible. The same situation Figure 4. Daily Tweet Emotions and Figure 5. Daily Sentiments. Drawings require refinement.
The conclusion is way too short. It is required to refer to the purpose of the work, the conducted research, and the results, but also to what literature shows activities in the researched area, what has been achieved.
Author Response
Good evening,
Herewith I attached the response. Thank you.

Round 2
Reviewer 1 Report
Thanks to authors for the revision. The paper is now acceptable.
Author Response
Thank you so very much for your help and support
Reviewer 2 Report
The article has been slightly improved, but there is still a lot to improve. The authors did not correct Figures 2-5 as recommended. It is still not known what dates appear on the X axis. The literature review still needs to be refined from the perspective of the latest studies from 2023. I recommend that the authors create separate results and discussion sections. Conclusion still pretty poor. The authors do not refer to the research results achieved or to what has been achieved in the literature in the field of the topic undertaken by the authors. The manuscript still needs a lot of work.
Author Response
Please find attached revision notes. We really appreciate all your help and support. Thank you.
